# Perception of sleep duration in adult patients with suspected obstructive sleep apnea

**Ricardo L. M. Duarte**[1,2], **Bruno A. Mendes**[3], **Tiago S. Oliveira-e-Sá**[3,4], **Flavio J. Magalhães-da-Silveira**[1], **David Gozal**[5]*

**1** SleepLab - Laboratório de Estudo dos Distúrbios do Sono, Rio de Janeiro, Brazil, **2** Instituto de Doenças do Tórax - Universidade Federal do Rio de Janeiro, Rio de Janeiro, Brazil, **3** Hospital de Santa Marta - Centro Hospitalar Universitário Lisboa Central, Lisbon, Portugal, **4** NOVA Medical School - Faculdade de Ciências Médicas, Universidade Nova de Lisboa, Lisbon, Portugal, **5** Department of Child Health and Child Health Research Institute, University of Missouri School of Medicine, Columbia, Missouri, United States of America

* gozald@health.missouri.edu

**Data Availability Statement:** All relevant data are within the paper and its Supporting Information files.

## Abstract

### Purpose

Discrepancies between subjective and objective measures of total sleep time (TST) are frequent among insomnia patients, but this issue remains scarcely investigated in obstructive sleep apnea (OSA). We aimed to evaluate if sleep perception is affected by the severity of OSA.

### Methods

We performed a 3-month cross-sectional study of Brazilian adults undergoing overnight polysomnography (PSG). TST was objectively assessed from PSG and by a self-reported questionnaire (subjective measurement). Sleep perception index (SPI) was defined by the ratio of subjective and objective values. Diagnosis of OSA was based on an apnea/hypopnea index (AHI) $\geq$ 5.0/h, being its severity classified according to AHI thresholds: 5.0–14.9/h (mild OSA), 15.0–29.9/h (moderate OSA), and $\geq$ 30.0/h (severe OSA).

### Results

Overall, 727 patients were included (58.0% males). A significant difference was found in SPI between non-OSA and OSA groups (p = 0.014). Mean SPI values significantly decreased as the OSA severity increased: without OSA (100.1 ± 40.9%), mild OSA (95.1 ± 24.6%), moderate OSA (93.5 ± 25.2%), and severe OSA (90.6 ± 28.2%), p = 0.036. Using logistic regression, increasing SPI was associated with a reduction in the likelihood of presenting any OSA (p = 0.018), moderate/severe OSA (p = 0.019), and severe OSA (p = 0.028). However, insomnia was not considered as an independent variable for the presence of any OSA, moderate/severe OSA, and severe OSA (all p-values > 0.05).

### Conclusion

In a clinical referral cohort, SPI significantly decreases with increasing OSA severity, but is not modified by the presence of insomnia symptoms.

**Funding:** The author(s) received no specific funding for this work.

**Competing interests:** The authors have declared that no competing interests exist.

## Introduction

Obstructive sleep apnea (OSA) is an extremely prevalent disorder [1], characterized by repetitive complete or partial airflow limitation due to increases in upper airway resistance during sleep, leading to intermittent hypoxemia and sleep fragmentation [2]. These hallmark characteristics of OSA are associated with cardiovascular, metabolic, and neurocognitive morbidities [2–5]. Diagnosis and assessment of OSA severity are based on the overnight polysomnographic study and routinely rely on the apnea-hypopnea index (AHI), which consists of the sum of apneas and hypopneas per hour of sleep. Although OSA is a frequent disease and is associated with high morbidity, it remains underdiagnosed [6, 7], especially due to the limited access, primarily in regions with scarce financial resources. Therefore, in adult individuals, portable home tests have quickly emerged as an alternative and effective method for the diagnosis of OSA, particularly in light of their reduced cost and wider availability when compared to full polysomnography (PSG) [8].

Another highly prevalent sleep disorder is chronic insomnia, and according to the definition used, insomnia rates can reach up to about 50% [9–11]. In general, the clinical diagnosis of insomnia is based on the presence of specific symptoms and their duration. Symptoms associated with chronic insomnia are difficulty in initiating sleep, difficulty maintaining sleep, and waking up earlier than desired, with one or more of such symptoms being present for at least 3 months [9–11]. Patients suffering from insomnia are more likely to use sleep medications, in addition to reporting more fragmented sleep and fewer total hours of sleep than those without a diagnosis of insomnia [9–11].

A mismatch between subjective data (patient complaints) and objective measurements collected from PSG may be related to fragmented sleep, which is more frequently present among patients with chronic insomnia than in patients with OSA and/or in the general population [12–22]. Among patients with insomnia, an overestimation of sleep latency and an underestimation of total sleep time (TST) are often observed [12–21]. In contrast, patients with OSA appear to have relative preservation of sleep perception (SP), whereas SP is worse in insomnia [21].

However, among symptomatic individuals being referred to the sleep laboratory for suspected OSA, a large cluster of comorbid OSA-insomnia is present [22–24], which may, therefore, change SP in OSA depending on the proportion of OSA-insomnia in the cohort. Studies focusing on SP concerning different levels of OSA severity are also scarce and available literature has focused on OSA as a dichotomous variable rather than explore the severity of OSA as a modifier. In the present study, we hypothesized that as OSA severity increases in clinically referred symptomatic patients, SP will be affected and that this effect may be independent of concurrent insomnia symptoms.

## Materials and methods

### Study design and patient selection

This cross-sectional study was carried out between December 2019 and February 2020. All participants were referred for sleep evaluation due to suspected sleep-disordered breathing by their attending physicians. Inclusion criteria consisted of individuals aged 18 years and older and with suspected OSA. Exclusion criteria were as follows: previously diagnosed OSA, use of home sleep studies for diagnosis, and incomplete clinical data concerning the absence of one or more of the following parameters: subjective analysis of TST, insomnia symptoms, use of sleeping medications, and/or Epworth Sleepiness Scale (ESS). Subsequently, eligible

individuals who underwent PSG and were diagnosed with central sleep apnea or those whose PSG was technically inadequate were also excluded from analyses.

## Ethics statement

The study protocol (#1.764.165) was approved by the Research Ethics Committee of the Federal University of Rio de Janeiro and was carried out by the ethical standards laid down in the 1964 Declaration of Helsinki and its later amendments. All participants gave written informed consent before any study procedure. The anonymity of each subject was strictly preserved.

## Clinical data acquisition

On the evening of the PSG and after collecting the consent form, gender, age, body mass index (BMI), neck circumference (NC), ESS, self-reported comorbidities (hypertension, diabetes mellitus, and insomnia), and regular use of sleep medications were systematically collected by experienced sleep technicians. BMI was calculated by dividing the weight in kilograms by the square of the height in meters ($kg/m^2$). NC (in cm) was measured using a flexible tape with all subjects in the upright sitting position, with the upper edge of the tape measure placed immediately below the laryngeal prominence and applied perpendicularly to the long axis of the neck. Subjective sleepiness was assessed using ESS, an 8-item questionnaire, with four-point scales [from zero to three] [25]. Score $\geq 11$ points (final score from 0 to 24 points) was considered as excessive daytime somnolence [25].

Chronic insomnia was defined as present if a patient indicated one or more of the following complaints, which were investigated through a semi-structured interview immediately before PSG: difficulty falling asleep, difficulty maintaining sleep, and/or waking up earlier than desired. Furthermore, these symptoms needed to occur at least 3 nights a week for $\geq 3$ months and to be related to the presence of functional daytime impairments [26].

## Overnight in-lab polysomnography

All sleep tests were conducted in a single-center: *SleepLab—Laboratório de Estudo dos Distúrbios do Sono*, Rio de Janeiro in Brazil. All participants underwent an attended, in-lab full PSG (EMBLA® S7000, Embla Systems, Inc., Broomfield, CO, USA) consisting of the recording of electroencephalography, electrooculography, electromyography (chin and legs), electrocardiography, airflow, thoracic and abdominal impedance belts, oxygen saturation, microphone for snoring, and body position sensors. Polysomnographic data were scored manually following the latest 2012 American Academy of Sleep Medicine guidelines [27] by two board-certified sleep physicians, who were blind to the results of the estimated TST measurements.

Obstructive apneas were defined as a decrease of at least 90% of airflow from baseline with persistent respiratory effort, lasting at least 10 seconds [27]. Hypopneas were classified as a reduction in the respiratory signal $\geq 30\%$ lasting $\geq 10$ seconds that were associated with $\geq 3\%$ oxygen desaturation or arousal [27]. The AHI was calculated as the combined number of apnea and hypopnea episodes per hour sleep. Polysomnographic diagnosis of OSA was based on AHI $\geq 5.0/h$, being its severity classified according to AHI thresholds: $\geq 5.0/h$ as any OSA, $\geq 15.0/h$ as moderate/severe OSA, and $\geq 30.0/h$ as severe OSA.

## Sleep perception index calculation

The next morning after PSG, all volunteers were asked to complete a questionnaire about their sleep study satisfaction and to estimate the subjective TST (min). (S1 File) Objective TST (min) was assessed from overnight PSG. Then, the sleep perception index (SPI) was calculated

as the ratio between the subjective TST estimate and the corresponding objective TST measurement [28].

## Statistical analysis

Data analysis was carried out using Statistical Package for the Social Sciences for Windows (SPSS; version 21.0; Chicago, IL, USA). Results are summarized as mean ± standard deviation or as number and percentage for continuous and categorical variables; respectively. Comparisons between groups were performed using the chi-squared test for categorical variables, while continuous variables were assessed using Student's *t*-test or univariate analysis of variance (ANOVA). Correlation was calculated through the Spearman's coefficient (r). Binary logistic regression was used to predict the relationship between predictors (SPI, presence of insomnia complaints, regular use of sleep-promoting medications, and ESS) and a dependent variable (AHI $\geq$ 5.0 *versus* < 5.0/h, $\geq$ 15.0 *versus* < 15.0/h, or $\geq$ 30.0 *versus* < 30.0/h). Estimates from logistic regression tests were expressed as odds ratios (OR) with the respective 95% confidence interval (CI). Calibration was evaluated by Hosmer-Lemeshow chi-squared test, being $p < 0.05$ considered as poor calibration [29]. Statistical tests were two-tailed and statistical significance was set at $p < 0.05$.

## Results

Of a total of 794 consecutive subjects referred for diagnostic PSG, 67 patients (8.4%) were subsequently excluded for the following reasons: 60 with incomplete clinical data, 5 tested with portable sleep studies, and 2 with previously diagnosed OSA. Therefore, a total of 727 adult individuals—including 422 males (58.0%) and 305 females (42.0%)—were considered eligible for further analyses. No PSG needed to be repeated.

Baseline patient characteristics are reported in Table 1. Overall, the mean age was 46.1 ± 16.2 years, mean BMI was 29.6 ± 5.9 kg/m$^2$, and mean NC was 39.2 ± 4.5 cm. As OSA severity increased, there was a proportional increase in the percentage of male individuals, in addition to increases in age, BMI, NC, presence of hypertension and diabetes, and excessive daytime sleepiness ($p < 0.001$ for all variables; Table 1). As would be expected in light of the inclusion of individuals with a high OSA pretest probability, there was a high frequency of any OSA (83.5%), moderate/severe OSA (58.5%), and severe OSA (35.5%).

### Sleep perception

Fig 1 illustrates the scatterplots between TST (subjective and objective measurements), SPI, and AHI. There was a positive correlation between subjective and objective values of TST ($p < 0.001$). Conversely, neither subjective TST nor objective TST was significantly associated with AHI: p-values: 0.063 and 0.226, respectively. Additionally, SPI was negatively correlated with AHI ($p = 0.006$).

As reported in Fig 2, mean SPI measurements were statistically different across OSA severity categories based on AHI: < 5.0/h, 5.0–14.9/h, 15.0–29.9/h, and $\geq$ 30.0/h; p = 0.036. Conversely, frequency of insomnia complaints was similar in patients classified according to OSA severity categories based on AHI thresholds: < 5.0/h (47.5%), 5.0–14.9/h (48.9%), 15.0–29.9/h (43.7%), and $\geq$ 30.0/h (41.9%), p = 0.464.

As shown in Fig 3, mean SPI values were statistically different according to three AHI cut-off points: < 5.0/h *versus* $\geq$ 5.0/h (p = 0.014); < 15.0/h *versus* $\geq$ 15.0/h (p = 0.018); and < 30.0/h *versus* $\geq$ 30.0/h (p = 0.026). On the other hand, frequency of insomnia symptoms was similar in patients classified according to these AHI cut-off points: < 5.0/h (47.5%)

**Table 1. Patient characteristics (n = 727).**

| Parameter | Total (n = 727) | Without OSA (n = 120) | Mild OSA (n = 182) | Moderate OSA (n = 167) | Severe OSA (n = 258) | p |
|---|---|---|---|---|---|---|
| **Clinical data** | | | | | | |
| Male gender, % | 422 (58.0) | 35 (29.2) | 94 (51.6) | 104 (62.3) | 189 (73.3) | < 0.001 |
| Age, years | 46.1 ± 16.2 | 36.1 ± 14.5 | 45.3 ± 15.4 | 46.6 ± 15.8 | 51.0 ± 15.5 | < 0.001 |
| BMI, kg/m$^2$ | 29.6 ± 5.9 | 26.9 ± 5.6 | 28.2 ± 5.6 | 30.1 ± 5.3 | 31.6 ± 6.0 | < 0.001 |
| NC, cm | 39.2 ± 4.5 | 35.9 ± 3.8 | 38.0 ± 3.8 | 39.5 ± 4.1 | 41.5 ± 4.1 | < 0.001 |
| ESS ≥ 11 points, % | 305 (42.0) | 40 (33.3) | 64 (35.2) | 67 (40.1) | 134 (51.9) | < 0.001 |
| Hypertension, % | 245 (33.7) | 18 (15.0) | 51 (28.0) | 53 (31.7) | 123 (47.7) | < 0.001 |
| Diabetes mellitus, % | 108 (14.9) | 5 (4.2) | 23 (12.6) | 26 (15.6) | 54 (20.9) | < 0.001 |
| Insomnia, % | 327 (45.0) | 57 (47.5) | 89 (48.9) | 73 (43.7) | 108 (41.9) | 0.464 |
| Sleep medications, % | 149 (20.5) | 25 (20.8) | 44 (24.2) | 38 (22.8) | 42 (16.3) | 0.182 |
| Subjective TST, min | 308.2 ± 97.3 | 303.8 ± 111.7 | 320.1 ± 96.0 | 316.2 ± 89.8 | 296.4 ± 94.8 | 0.055 |
| **Polysomnographic data** | | | | | | |
| Objective TST, min | 334.8 ± 74.7 | 314.7 ± 80.9 | 339.6 ± 75.1 | 342.9 ± 65.7 | 335.5 ± 75.5 | 0.009 |
| Sleep efficiency, % | 77.5 ± 37.3 | 73.2 ± 17.4 | 77.6 ± 15.9 | 83.3 ± 71.8 | 75.6 ± 16.0 | 0.097 |
| Arousal index, n/h | 29.9 ± 23.7 | 6.7 ± 3.0 | 13.5 ± 4.5 | 24.1 ± 4.8 | 56.0 ± 20.3 | < 0.001 |
| AHI, n/h | 27.6 ± 25.4 | 2.1 ± 1.3 | 9.7 ± 3.0 | 21.2 ± 4.1 | 56.3 ± 20.5 | < 0.001 |
| Mean SpO$_2$, % | 93.6 ± 2.3 | 95.2 ± 1.5 | 94.2 ± 1.7 | 93.8 ± 1.9 | 92.3 ± 2.5 | < 0.001 |
| Lowest SpO$_2$, % | 82.5 ± 8.4 | 88.0 ± 7.4 | 85.0 ± 6.4 | 84.4 ± 5.2 | 77.1 ± 8.8 | < 0.001 |
| ODI at 3%, n/h | 19.0 ± 23.0 | 1.4 ± 1.6 | 5.3 ± 5.1 | 12.1 ± 7.4 | 41.6 ± 25.1 | < 0.001 |

Data were presented as mean ± standard deviation or n (%). BMI: body-mass index; NC: neck circumference; ESS: Epworth Sleepiness Scale; TST: total sleep time; AHI: apnea/hypopnea index; SpO$_2$: oxygen saturation; ODI: oxygen desaturation index. TST was objectively evaluated from PSG, while its subjective measure was filled out using a questionnaire. Polysomnographic diagnosis of obstructive sleep apnea (OSA) was based on AHI thresholds: < 5.0/h (without OSA), 5.0–14.9/h (mild OSA), 15.0–29.9/h (moderate OSA), and ≥ 30.0/h (severe OSA).

versus ≥ 5.0/h (44.5%), p = 0.549; < 15.0/h (48.3%) *versus* ≥ 15.0/h (42.6%), p = 0.131; and < 30.0/h (46.7%) *versus* ≥ 30.0/h (41.9%), p = 0.214.

Of note, only males showed a statistically significant reduction in mean SPI values as the severity of OSA increased: SPI ranged from 109.3 ± 50.5% (without OSA) to 88.1 ± 26.6% (severe OSA), p < 0.001. In females, however, SPI was similar concerning increasing AHI-based categories, ranging from 96.3 ± 36.0% (without OSA) to 97.8 ± 31.4% (severe OSA), p = 0.935.

## Binary logistic regression

Table 2 shows the logistic regression that was performed to verify the effects of SPI, insomnia symptoms, regular use of sleep-promoting medications, and ESS on the likelihood of individuals having any OSA, moderate/severe OSA, and severe OSA. Increasing SPI was associated with a reduction in the likelihood of presenting any OSA (OR: 0.992 [95% CI: 0.986–0.999]; p = 0.018), moderate/severe OSA (OR: 0.994 [95% CI: 0.988–0.999]; p = 0.019), and severe OSA (OR: 0.994 [95% CI 0.988–0.999]; p = 0.028). Increasing the ESS score was associated with a higher probability of exhibiting moderate/severe OSA (p = 0.010) and severe OSA (p < 0.001). Also, neither the presence of insomnia nor the regular use of sleep-promoting medications was considered as an independent variable for the presence of any OSA, moderate/severe OSA, and severe OSA (all p-values > 0.05).

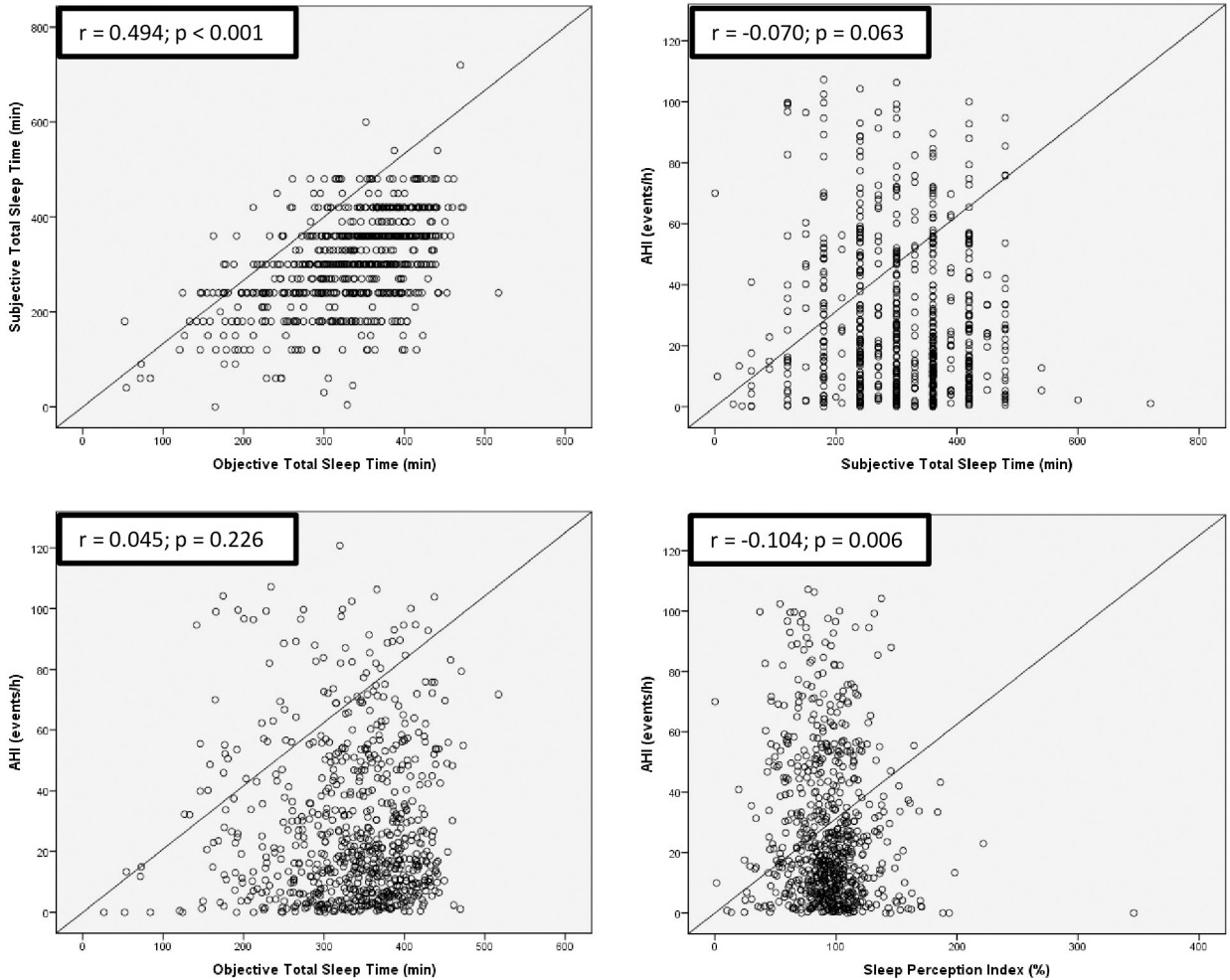

**Fig 1. Scatterplot of Apnea/Hypopnea Index (AHI), subjective and objective Total Sleep Time (TST), and Sleep Perception Index (SPI) in 727 individuals.** Subjective TST value was self-reported by participants, while objective TST value was measured from full polysomnography. SPI was defined as the ratio between the subjective and objective TST. Spearman's rank correlation coefficient (r) was used to evaluate the strength of association between two variables.

## Discussion

The main findings of our study were that SPI values progressively decreased with increasing severity of OSA, indicating that individuals with more severe forms of OSA tend to have a worse SP. It is therefore likely that the increasing severity of OSA manifesting both as more severe intermittent hypoxemia (increased oxygen desaturation index) and sleep fragmentation (increased arousal index) may induce progressive sleep satisfaction reductions [21, 30–32]. We further explore whether concurrent insomnia-related symptoms could have affected our findings since individuals with insomnia noticeably have a lower SPI when compared with OSA patients or controls [12–20]. As the frequency of insomnia complaints was similar across the AHI categorical thresholds (5.0/h, 15.0/h, and 30.0/h), the presence or absence of insomnia is highly unlikely to have biased our findings. Moreover, this parameter was not considered as an independent variable for the diagnosis of OSA, regardless of its severity (all p-values > 0.05).

Unlike our findings, a previous study that included 248 individuals (42% without OSA and 58% with OSA) reported no differences in the objective and subjective measures of TST in

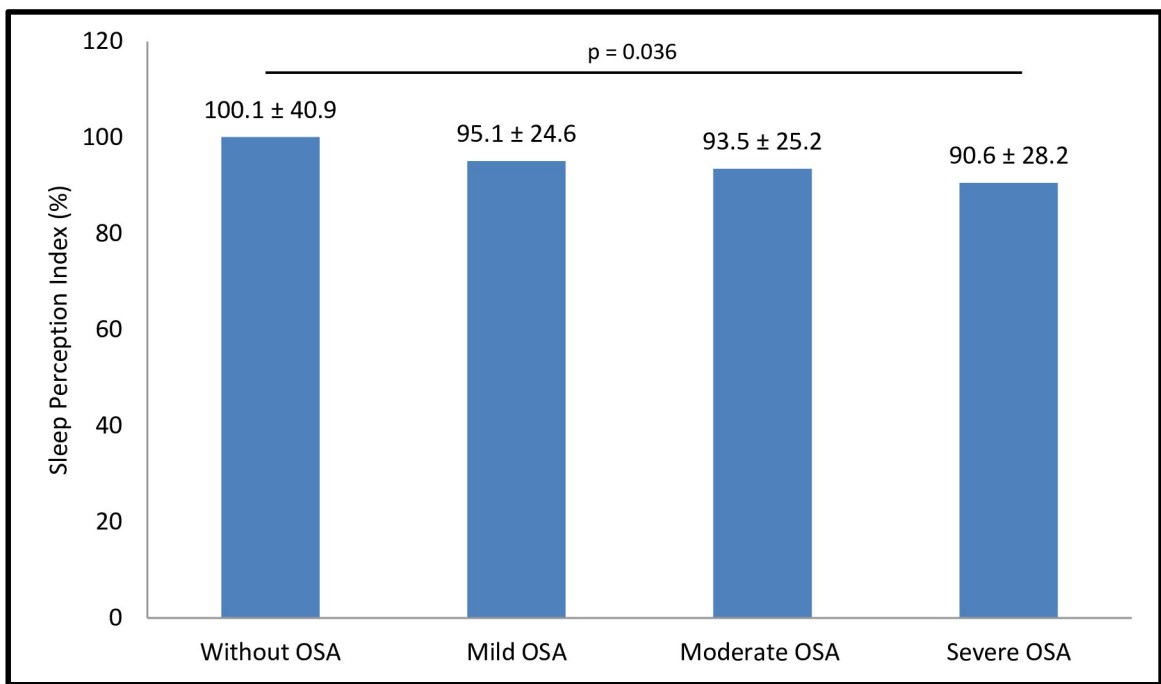

**Fig 2. Sleep Perception Index (SPI) and Obstructive Sleep Apnea (OSA) severity.** SPI measurements were statistically different across categories of OSA severities: < 5.0/h (without OSA), 5.0–14.9/h (mild OSA), 15.0–29.9/h (moderate OSA), and ≥ 30.0/h (severe OSA). SPI was defined as the ratio between the subjective and objective total sleep time. Values were reported as mean ± standard deviation.

patients diagnosed with OSA [20]. However, this study defined only one AHI threshold at 5.0/h to categorize patients with or without OSA, while we expanded the analysis to a larger cohort of subjects and applied the three AHI thresholds most commonly used to assess OSA severity. This approach revealed that SP was significantly reduced with the concomitant increase in OSA severity, but found no significant association with the frequency of comorbid OSA-insomnia.

Another study postulated that compared with patients diagnosed with mild OSA, those diagnosed with severe OSA would exhibit greater sleep fragmentation, which in turn could lead to a worse SP [32]. This study enrolled 50 OSA patients which included 30 with normal SP and 20 with abnormal SP, i.e., perceived TST < 80% of TST measured in PSG [32]. No statistically significant difference in SP emerged based on different OSA severities: 0.75 ± 0.21 (mild OSA), 0.89 ± 0.18 (moderate OSA), and 0.82 ± 0.20 (severe OSA); p = 0.19 (32). However, the small size sample, the use of the 2007 American Academy of Sleep Medicine scoring criteria, and the absence of controls (subjects without OSA) are all substantial limitations that may account for the negative findings [32].

In our study, men, but not women, showed a statistically significant reduction in mean SPI measurements as the severity of OSA increased. This finding, although intriguing, should be extrapolated to the general population with caution, since men are more likely to suffer OSA than women, while women typically are more likely to present with increased insomnia symptoms when compared to men. Moreover, other covariates, such as age and BMI, should also be concurrently evaluated to establish whether there are SP differences between men and women, or whether these are only the result of the enrolled population.

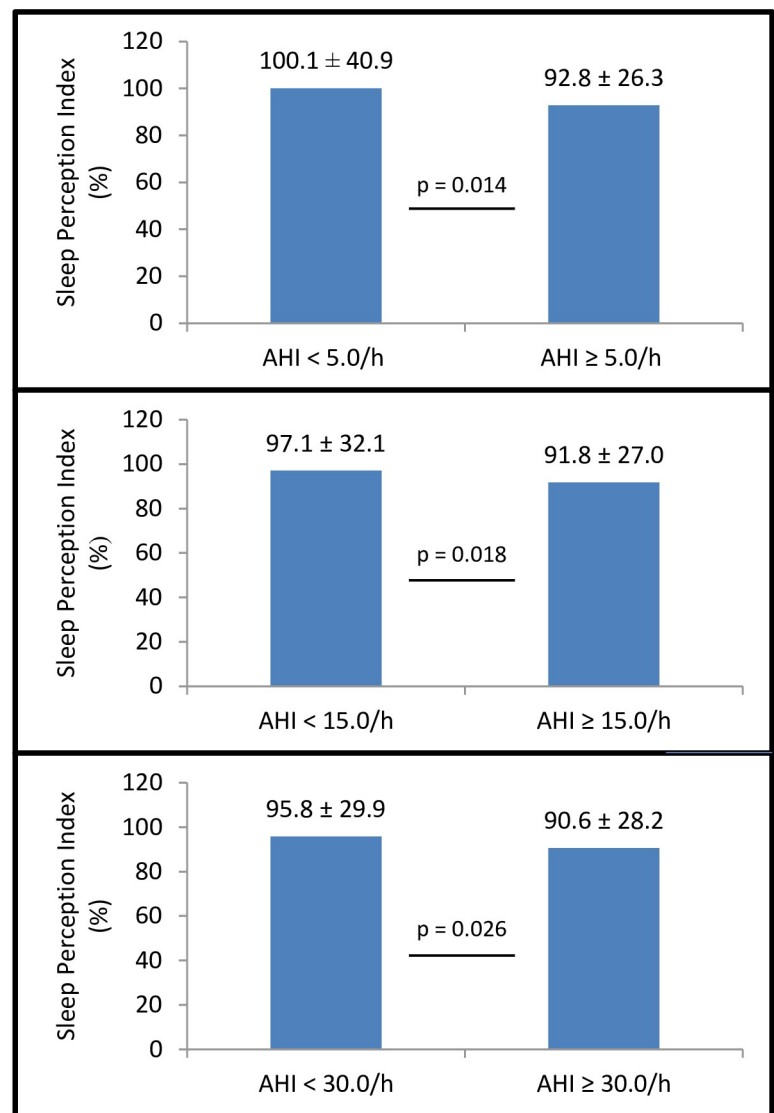

**Fig 3. Sleep Perception Index (SPI) and Apnea/Hypopnea Index (AHI) thresholds.** SPI measurements were statistically different according to three AHI thresholds: $< 5.0/h$ *versus* $\geq 5.0/h$, $< 15.0/h$ *versus* $\geq 15.0/h$, and $< 30.0$ *versus* $\geq 30.0/h$. SPI was defined as the ratio between the subjective and objective total sleep time. Values were shown as mean ± standard deviation.

## Limitations and strengths

Our study had some limitations that deserve mention. All participants were referred to a single sleep laboratory, which may limit the reproducibility and generalization of our findings. Other measures of SP commonly evaluated mainly in individuals suffering from insomnia, such as sleep latency and awakenings, were not included, but since the prevalence of insomnia was similar across OSA severity groups, the impact of such omission should be minimal if any. Another possible limitation was that the objective measurement of TST was extracted exclusively from a single night in the sleep laboratory; therefore, night-to-night variability or the effects of the first night could not be completely excluded. However, the inclusion of a relatively robust sample of consecutive adult individuals who underwent full in-lab PSGs is an

**Table 2. Binary logistic regression of covariates according to AHI thresholds (n = 727).**

| | β | SE | Wald | df | p-value | OR (95% CI) |
|---|---|---|---|---|---|---|
| **AHI ≥ 5.0/h** | | | | | | |
| Sleep perception index | -0.008 | 0.003 | 5.583 | 1 | 0.018 | 0.992 (0.986–0.999) |
| Insomnia complaints | -0.125 | 0.212 | 0.349 | 1 | 0.555 | 0.882 (0.583–1.336) |
| Sleep medications use | 0.085 | 0.261 | 0.106 | 1 | 0.745 | 1.089 (0.653–1.816) |
| ESS score | 0.036 | 0.021 | 2.960 | 1 | 0.085 | 1.037 (0.995–1.081) |
| **AHI ≥ 15.0/h** | | | | | | |
| Sleep perception index | -0.006 | 0.003 | 5.500 | 1 | 0.019 | 0.994 (0.988–0.999) |
| Insomnia complaints | -0.213 | 0.161 | 1.753 | 1 | 0.185 | 0.808 (0.589–1.108) |
| Sleep medications use | -0.087 | 0.198 | 0.192 | 1 | 0.661 | 0.917 (0.623–1.351) |
| ESS score | 0.041 | 0.016 | 6.656 | 1 | 0.010 | 1.042 (1.010–1.074) |
| **AHI ≥ 30.0/h** | | | | | | |
| Sleep perception index | -0.006 | 0.003 | 4.854 | 1 | 0.028 | 0.994 (0.988–0.999) |
| Insomnia complaints | -0.176 | 0.168 | 1.093 | 1 | 0.296 | 0.839 (0.603–1.166) |
| Sleep medications use | -0.193 | 0.214 | 0.812 | 1 | 0.368 | 0.825 (0.542–1.254) |
| ESS score | 0.060 | 0.016 | 13.448 | 1 | < 0.001 | 1.062 (1.028–1.097) |

AHI: apnea/hypopnea index; ESS: Epworth Sleepiness Scale, β: regression coefficient, SE: standard error, df: degrees of freedom for the Wald test, OR: odds ratio, CI: confidence interval. Sleep perception index was defined as the ratio between the subjective and objective total sleep time. The logistic model regression model showed adequate calibration, which was accessed by Hosmer–Lemeshow test: any OSA (AHI ≥ 5.0/h): 8.987 (p = 0.343); moderate/severe OSA (AHI ≥ 15.0/h): 14.572 (p = 0.068); and severe OSA (AHI ≥ 30.0/h): 4.754 (p = 0.783).

obvious strength of the present study. Moreover, both physicians responsible for PSG reports were blinded to the subjective TST estimate values.

## Conclusions

In symptomatic adults who are clinically referred to a sleep laboratory, SPI was lower in those subjects with OSA diagnosis. Furthermore, among those diagnosed with OSA, SPI decreased according to the severity of OSA increased, and this relationship was not influenced by the presence of clinical symptoms of insomnia.

## Supporting information

**S1 File.**
(DOCX)

## Author Contributions

**Conceptualization:** Ricardo L. M. Duarte, Bruno A. Mendes, Flavio J. Magalhães-da-Silveira.

**Data curation:** Ricardo L. M. Duarte, Bruno A. Mendes.

**Formal analysis:** Ricardo L. M. Duarte.

**Investigation:** Ricardo L. M. Duarte, Bruno A. Mendes.

**Methodology:** Ricardo L. M. Duarte, Bruno A. Mendes, Tiago S. Oliveira-e-Sá, Flavio J. Magalhães-da-Silveira, David Gozal.

**Project administration:** Flavio J. Magalhães-da-Silveira, David Gozal.

**Writing – original draft:** Ricardo L. M. Duarte.

**Writing – review & editing:** Ricardo L. M. Duarte, Bruno A. Mendes, Tiago S. Oliveira-e-Sá, Flavio J. Magalhães-da-Silveira, David Gozal.

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
