## [Decision Letter · Decision Letter 0]

14 Jul 2020

PONE-D-20-18917

Perception of sleep duration in adult patients with suspected obstructive sleep apnea

PLOS ONE

Dear Dr. Gozal,

Thank you for submitting your manuscript to PLOS ONE. After careful consideration, we feel that it has merit but does not fully meet PLOS ONE’s publication criteria as it currently stands. Therefore, we invite you to submit a revised version of the manuscript that addresses the points raised during the review process.

The academic editor agrees with the reviewers comments and hopes that the authors can reply to them in detail.

We look forward to receiving your revised manuscript.

Kind regards,

James Andrew Rowley

Academic Editor

PLOS ONE

Journal Requirements:

2. Please include additional information regarding the questionnaire about their sleep study satisfaction used in the study and ensure that you have provided sufficient details that others could replicate the analyses. For instance, if the questionnaire it is not under a copyright more restrictive than CC-BY, please include a copy, in both the original language and English, as Supporting Information.

Reviewers' comments:

Reviewer's Responses to Questions

**Comments to the Author**

1. Is the manuscript technically sound, and do the data support the conclusions?

Reviewer #1: Yes

Reviewer #2: Yes

2. Has the statistical analysis been performed appropriately and rigorously? 

Reviewer #1: Yes

Reviewer #2: Yes

3. Have the authors made all data underlying the findings in their manuscript fully available?

Reviewer #1: Yes

Reviewer #2: Yes

4. Is the manuscript presented in an intelligible fashion and written in standard English?

Reviewer #1: Yes

Reviewer #2: Yes

5. Review Comments to the Author

Reviewer #1: Discrepancies between subjective and objective measures of total sleep time are frequent among insomnia patients, but this issue remains scarcely investigated in OSA. The authors attempted to evaluate if sleep perception is affected by the severity of OSA. They performed a 3-month cross-sectional study of Brazilian adults undergoing PSG. Objective sleep time was assessed from PSG and subjective sleep time by a self-reported questionnaire. Sleep perception index was defined by the ratio of subjective and objective values.Overall, 727 patients were included (58.0% males). A significant difference was found in SPI between non-OSA and OSA groups, and SPI significantly decreases with increasing OSA severity.

This was a very interesting and well-written paper or an important topic in sleep medicine. My comments are minor in nature. For example, in the 2nd paragraph of Introduction, the authors should specify that this is "chronic" insomnia (i.e. > 3 mos). Acute insomnia is quite different in nature and may be confusing to some readers. Additionally, on line 190, it should say "TST" not TTS. Otherwise, no major issues, the authors do a great job of listing Limitations.

Reviewer #2: This is an interesting paper that reflects a phenomenon often observed in clinical practice.

Introduction:

- Would also mention the role of HSATs in OSA diagnosis.

- Say "patients with insomnia" rather than "insomniacs"

- Appreciate that the hypothesis is clearly stated

Methods:

- Please clarify what is meant by "incomplete clinical data" as an exclusion criterion

- Why was information about medical comorbidities collected by self-report rather than by chart review?

- Were there particularly categories of "sleep medications" that were inquired about for this study? Information collected on duration of medication use?

- Please explain why GOAL was used as the screening questionnaire (rather than other, more commonly encountered questionnaires)

Results:

- Under "Sleep Perception" - please define TTS

- Why was the discriminatory ability of the ESS and GOAL questionnaire determined? This does not add new information to the literature, and was not stated as a hypothesis/aim for the current study.

Discussion:

- Please clarify - are you defining sleep fragmentation increased oxygen desaturation index and arousal index? If so, please cite references for both components of this definition.

- The paragraphs about the GOAL questionnaire and ESS are unrelated to the rest of the manuscript and stated hypothesis/aim; suggest to remove these paragraphs.

- Please include further discussion about the gender discrepancies and SPI mentioned in the Results section.

6. PLOS authors have the option to publish the peer review history of their article (what does this mean?). If published, this will include your full peer review and any attached files.

Reviewer #1: **Yes: **Daniel A. Barone, MD

Reviewer #2: No

---

## [Author Response · Author response to Decision Letter 0]

24 Jul 2020

Dear Editor and Reviewers:

We would like to thank the reviewers for their comments and acknowledge the contributions of such critiques to improvements in the manuscript. The detailed answer to all the pertinent questions raised by the Reviewers follows below. 

5. Review Comments to the Author

Reviewer #1: Discrepancies between subjective and objective measures of total sleep time are frequent among insomnia patients, but this issue remains scarcely investigated in OSA. The authors attempted to evaluate if sleep perception is affected by the severity of OSA. They performed a 3-month cross-sectional study of Brazilian adults undergoing PSG. Objective sleep time was assessed from PSG and subjective sleep time by a self-reported questionnaire. Sleep perception index was defined by the ratio of subjective and objective values. Overall, 727 patients were included (58.0% males). A significant difference was found in SPI between non-OSA and OSA groups, and SPI significantly decreases with increasing OSA severity.

This was a very interesting and well-written paper or an important topic in sleep medicine. My comments are minor in nature. 

For example, in the 2nd paragraph of Introduction, the authors should specify that this is "chronic" insomnia (i.e. > 3 mos). Acute insomnia is quite different in nature and may be confusing to some readers. 

Following the Reviewer's comment, we now use the term chronic insomnia to better characterize this disorder.

Additionally, on line 190, it should say "TST" not TTS. Otherwise, no major issues, the authors do a great job of listing Limitations.

We apologize for the oversight which is now corrected. 

Reviewer #2: This is an interesting paper that reflects a phenomenon often observed in clinical practice.

Introduction:

- Would also mention the role of HSATs in OSA diagnosis.

As suggested by the Reviewer, the introduction was rewritten mentioning the role of home sleep studies in the diagnosis of OSA.

- Say "patients with insomnia" rather than "insomniacs"

Accordingly, the term “insomniacs” was has been replaced by “patients with insomnia”.

- Appreciate that the hypothesis is clearly stated

Methods:

- Please clarify what is meant by "incomplete clinical data" as an exclusion criterion

The “incomplete clinical data” mention refers to the absence of one or more of the following clinical parameters of interest to the study: subjective analysis of total sleep time, insomnia symptoms, use of sleeping medications, and/or Epworth Sleepiness Scale. An explanation has now been included in the Methods section.

- Why was information about medical comorbidities collected by self-report rather than by chart review? 

Comorbidities (hypertension, diabetes mellitus, and insomnia symptoms) were systematically collected immediately before the polysomnography was performed, thus it was not necessary to resort to medical record review.

- Were there particularly categories of "sleep medications" that were inquired about for this study? Information collected on duration of medication use?

Unfortunately, data related to the different types of sleeping medications were not systematically collected. Instead, the use of sleeping pills was categorically answered as yes or no. However, it should be emphasized that only the regular/daily use of these drugs was considered to be a positive response. This observation on the regular use of these drugs has now been placed in the revised version of the manuscript.

- Please explain why GOAL was used as the screening questionnaire (rather than other, more commonly encountered questionnaires)

The GOAL questionnaire is a simplified instrument for OSA screening in adults, which was recently published (2020), possibly explaining why it is less cited than other instruments with a similar purpose. Notwithstanding, we agree with the Reviewer that its use is outside the main scope of the study, justifying removal.

Results:

- Under "Sleep Perception" - please define TTS

We apologize for the spelling mistake being the correct term (TST) inserted in the manuscript.

- Why was the discriminatory ability of the ESS and GOAL questionnaire determined? This does not add new information to the literature, and was not stated as a hypothesis/aim for the current study.

As explained above and according to the Reviewer's comment, we removed the data regarding the discriminatory power of the GOAL and ESS, since these data are outside the main objectives of the study.

Discussion:

- Please clarify - are you defining sleep fragmentation increased oxygen desaturation index and arousal index? If so, please cite references for both components of this definition.

We agree with the Reviewer that the paragraph deserves further clarification. We rewrote the text explaining that intermittent hypoxemia is related to the oxygen desaturation index, while sleep fragmentation is related to the arousal index.

- The paragraphs about the GOAL questionnaire and ESS are unrelated to the rest of the manuscript and stated hypothesis/aim; suggest to remove these paragraphs.

As mentioned above, these paragraphs have been removed from the revised manuscript.

- Please include further discussion about the gender discrepancies and SPI mentioned in the Results section.

As insightfully suggested by the Reviewer, we have written a new paragraph with possible gender-related discrepancies about sleep perception. 

Sincerely,

The Authors

---

## [Decision Letter · Decision Letter 1]

10 Aug 2020

Perception of sleep duration in adult patients with suspected obstructive sleep apnea

PONE-D-20-18917R1

Dear Dr. Gozal,

We’re pleased to inform you that your manuscript has been judged scientifically suitable for publication and will be formally accepted for publication once it meets all outstanding technical requirements.

Kind regards,

James Andrew Rowley

Academic Editor

PLOS ONE

Additional Editor Comments (optional):

Reviewers' comments:

Reviewer's Responses to Questions

**Comments to the Author**

1. If the authors have adequately addressed your comments raised in a previous round of review and you feel that this manuscript is now acceptable for publication, you may indicate that here to bypass the “Comments to the Author” section, enter your conflict of interest statement in the “Confidential to Editor” section, and submit your "Accept" recommendation.

Reviewer #1: All comments have been addressed

Reviewer #2: All comments have been addressed

2. Is the manuscript technically sound, and do the data support the conclusions?

Reviewer #1: Yes

Reviewer #2: Yes

3. Has the statistical analysis been performed appropriately and rigorously? 

Reviewer #1: Yes

Reviewer #2: I Don't Know

4. Have the authors made all data underlying the findings in their manuscript fully available?

Reviewer #1: Yes

Reviewer #2: Yes

5. Is the manuscript presented in an intelligible fashion and written in standard English?

Reviewer #1: Yes

Reviewer #2: Yes

6. Review Comments to the Author

Reviewer #1: The authors did a fine job and have addressed all my (minor) concerns. From my perspective, the paper is ready to be published.

Reviewer #2: Thanks for the opportunity to review this manuscript revision.

The authors have addressed all initially stated concerns in this revision.

7. PLOS authors have the option to publish the peer review history of their article (what does this mean?). If published, this will include your full peer review and any attached files.

Reviewer #1: **Yes: **Daniel A Barone

Reviewer #2: No

---

## [Editor Report · Acceptance letter]

17 Aug 2020

PONE-D-20-18917R1 

Perception of sleep duration in adult patients with suspected obstructive sleep apnea 

Dear Dr. Gozal:

I'm pleased to inform you that your manuscript has been deemed suitable for publication in PLOS ONE. Congratulations! Your manuscript is now with our production department. 

Kind regards, 

on behalf of

Dr. James Andrew Rowley 

Academic Editor

PLOS ONE